



# Exploring spatial-temporal dynamics of fire regime features at mainland Spain

Adrián Jiménez-Ruano[1], Marcos Rodrigues Mimbrero[1], Juan de la Riva Fernández[1]

[1]GEOFOREST Group, IUCA, Department of Geography and Land Management, University of Zaragoza,

Pedro Cerbuna 12, 50009, Zaragoza, Spain

*Correspondence to:* Adrián Jiménez-Ruano (jimenez@unizar.es)

**Abstract**

This paper explores spatial-temporal dynamics in fire regime features, such as fire frequency, burnt area,

large fires, and natural- and human-caused fires, as an essential part of fire regime characterisation. Changes in fire features are analysed at different spatial – regional and provincial/NUTS3 levels, together with summer and winter temporal scales, using historical fire data from Spain for the period 1974-2013. Temporal shifts in fire features are investigated by means of change point detection procedures – Pettitt test, AMOC (At Most One Change), PELT (Pruned Exact Linear Time) and BinSeg (Binary

Segmentation) – at regional level to identify changes in the time series of the features. A trend analysis was conducted using the Mann-Kendall and Sen's slope tests at both regional and NUTS3 level. Finally, we applied a Principal Component Analysis (PCA) and Varimax Rotation to trend outputs – mainly Sen's slope values – to summarize overall temporal behaviour, also to explore potential links in the evolution of fire features.

Our results suggest that most fire features show remarkable shifts between the late 1980s and the first half of the 1990s. Mann-Kendall outputs revealed negative trends in the Mediterranean region. Results from Sen's slope suggest high spatial and intra-annual variability across the study area. Fire activity related to human sources seems to be experiencing an overall decrease in the north-west provinces, particularly pronounced during summer. Conversely, the hinterlands and the Mediterranean coast are gradually

becoming less fire-affected. Finally, PCA enabled trends to be synthesized into four main components: winter fire frequency (PC1), summer burnt area (PC2), large fires (PC3) and natural fires (PC4).






## 1 Introduction

Wildfire is a disturbance affecting many ecosystems on a global level. Fire itself is a very dynamic
landscape process, which depends on different factors, such as weather, vegetation type, fuel moisture,
land use or human activity, among others (Falk et al., 2011). Understanding wildfire phenomenon is still a
challenging task, especially difficult when facing climate and/or socioeconomic changes (Pausas and
Keeley, 2009) as is the case of Spain (Pausas, 2004; Pausas and Fernández-Muñoz, 2012; Turco et al.,

2014) and other EUMed regions (Moriondo et al., 2006; Salis et al., 2014; Venäläinen et al., 2014). In this
context, fire regime characterisation may contribute to improving our knowledge on how wildfire works.
For example, understanding spatial and temporal patterns of wildfire features may lead to more effective
management strategies or prevention policies.

Improved understanding of fire regimes may be achieved by providing deeper insights into the spatial

patterns of fire regime features over a certain period of time (i.e. homogenous areas with similar fire
regime characteristics or pyroregions). From a temporal perspective, previous studies reported the
existence of temporal changes or trends in the evolution of several fire regime features (Flannigan et al.,
2009; Rodrigues et al., 2013; Turco et al., 2016). However, despite there being evidence of temporal
variability in fire regime features it is not usually considered when defining, characterizing or outlining

fire regimes. The concept of fire regime is commonly defined as the average conditions of fire that remain
recurrent and consistent within a particular area and occurring over a certain period of time (Krebs et al.,
2010). According to this definition, it seems clear that both space and time are taken to be stationary, but
fire regime components are in fact highly variable across time and space (Morgan et al., 2001). Recently,
several works studying fire regimes were conducted in Spain, among which those by Moreno and

Chuvieco (2012) and Moreno and Chuvieco (2016) are notable as the latest attempts to deal with fire
regime characterisation. However, in these works, the behaviour of fire regime features is still assumed to
be homogeneous or stationary over time. To our understanding, the concept or definition of fire regime
has to include changes in fire features over the study period. This is ultimately the goal of our proposal, to
characterise the temporal evolution of fire regime features so that they can be employed to refine and

improve the spatial outline of fire regimes.

Analyses of spatial-temporal trends of fire regime features are common in the literature. The most
widespread approach addresses changes in fire frequency and burned area (Kasischke and Turetsky,
2006; Pausas and Fernández-Muñoz, 2012; Rodrigues et al., 2013; Zavala et al., 2011). In the
Mediterranean region, the main findings indicate a general decrease during the period 1985-2011 (Turco

et al., 2016) in the annual number of wildfires and burned area, although a certain spatial variability is
observed in the trends. For instance, over the last few decades, the burned area in Spain has decreased.
Conversely, the yearly number of fires have increased, except on the Mediterranean coast (Turco et al.,
2016). Since most studies focus mainly on analysing 'generic' fire (number of fires and burned area).
This is particularly true for Spain, which lacks a detailed analysis of fire trends based on a spatial-

temporal approach. We believe that a proper characterisation of fire regime must take into account
additional features, such as fire size, cause or seasonality. Even though some studies dealing with the
temporal dimension of wildfires do exist (Serra et al., 2013), most of them present some limitations, such





as a short time series (less than 20 years of data). Meanwhile, analyses using a longer time series do not include many fire regime features and stay with the overall number of fires and burned area (Pausas 2004; Pausas and Fernández-Muñoz 2012; Moreno et al. 2014). Specifically, we stress the importance of assessing the evolution of large fires (fires with more than 500 ha burned, San-Miguel-Ayanz et al. (2013), and the potential differences relating the ignition source of a wildfire, either natural- or human-caused.

Therefore, the analysis of the temporal dimension of fire regime features must be extended to these other features in order to provide a more detailed picture of the evolution of fire activity with the ultimate goal of characterizing fire regimes. Similarly, advances must be made in applying trend detection procedures. Determining whether a certain feature changes significantly is useful, but not sufficient. Further insights into trend magnitude must be provided so that we can compare trends among several regions, and explore possible relationships among temporal changes in fire features.

The aim of this study is to analyse spatial-temporal trends of several fire features during the period 1974-2013, and explore potential relationships among those detected. The analysis is conducted at several spatial, such as regional and NUTS3/provincial level, and autumn-winter and spring-summer temporal – scales in mainland Spain. Data on fire regime features were retrieved from historical fire records stored in the EGIF (General Wildfires Statistics) database. Firstly, seasonal shifts in the evolution of fire feature were examined using change point detection techniques in the Northwest, Hinterland and Mediterranean pyroregions comprising the whole of mainland Spain –. We used a Pettitt test, AMOC (At Most One Change), PELT (Pruned Exact Linear Time) and BinSeg (Binary Segmentation) to determine if and when significant changes in the temporal evolution of each fire regime feature takes place. Trend detection procedures were then applied at regional (Mann-Kendall) and NUTS3 level (Mann-Kendall and Sen`s Slope), to address the spatial-temporal variability of each fire feature. The purpose was to determine whether fire features vary over time and, if so, its sign – upward or downward – and strength. Finally, a Principal Component Analysis (PCA) coupled to Varimax Rotation was applied to uncover potential links in the evolution of fire features.

## 2 Materials and methods

### 2.1 Study area

The study area encompasses the whole of mainland Spain (thus excluding both the Balearic and Canary archipelagos, as well as the autonomous cities of Ceuta and Melilla). Spain is very biophysically diverse, presenting a wide variety of climatic, topographical, and environmental conditions. This diversity also appears when discussing socioeconomic conditions, in terms of settlement systems and population structure, production sector, changes in land use and land cover, or structure of the territory.

From the biogeographical standpoint, mainland Spain is dominated by two regions: Eurosiberian and Mediterranean. The Eurosiberian region covers most of the northern area of the country. It is characterized by an Oceanic climate (according to Koeppen's climate classification - *Cfb*), mostly covered by various types of vegetation from deciduous oak and ash to evergreen oak woodlands (*Quercus*





*robur, Fraxinus excelsior* or *Fagus sylvatica*). The Mediterranean region covers the remaining territory. Hot-summer Mediterranean (*Csa*) and cold semi-arid (*BSk*) climates characterize this area, which therefore has notably drier and warmer conditions than the Eurosiberian region. These conditions, coupled to human activity, favour complex mosaics of agricultural systems and plant communities. Sclerophyllous and evergreen vegetation, such as *Quercus ilex* and thermophilous scrublands (maquis and garrigues formations), dominate the region, and forest areas mainly consist of pines (*Pinus halepensis*). Furthermore, bioclimatic (altitudinal) belts exist within each region in mountain areas such as the Pyrenees along the French border or Sierra Nevada on the southern Mediterranean coast.

Due to the variety of landscapes, climate and socioeconomic conditions, three different pyroregions – Northwest (NW), Hinterland (HL) and Mediterranean (MED) – were outlined (Fig.1), following the criteria established by the Spanish Environmental Ministry in their annual fire reports. These regions portray homogeneous areas in terms of wildfire activity by merging entire provinces or Autonomous Communities. The NW region includes the Autonomous Communities of Galicia, Asturias, Cantabria and the Basque Country, also the provinces of León and Zamora. This region is located within the Eurosiberian region, excluding the Pyrenees. The HL region includes all of the Autonomous Communities without coastline, except for the provinces of León and Zamora (included in the NW region). This region is located in the transition boundary between the Mediterranean and Eurosiberian regions, thus sharing characteristics in terms of climate influence and plant species. Finally, the MED region, situated within the Mediterranean biogeographical region, includes all the Autonomous Communities along the Mediterranean coast.

Figure 1

### 2.2 Fire data

Fire records from 1974 to 2013 were collected from the EGIF database and fire count data, the size of the total burned area, ignition triggering date and fire cause were retrieved for each fire event, later summarized by season at NUTS3 level. Only information on fires larger than 1 ha was retained because small fires (i.e. fires with less than 1 ha affected) were not fully compiled until 1988. Additionally, it is important to remember that in the case of the Autonomous Community of Navarre, fire data were only available from 1988. Hence, all the analyses conducted in Navarre were based on a slightly different study period (1988-2013).

In addition to NUTS3 level, we also included a regional scale of aggregation, together with two different fire seasons. Thus, annual data were split into spring-summer season (S), from April to September; and autumn-winter season (W) from October to March. From all available fire data information, eight fire features were then constructed for each fire season, NUTS and region:

- Fire frequency (N): total number of fires, regardless size or ignition source.
- Burned area size (B): total fire affected area, regardless size or ignition source.





- Number of large fires (N500): number of fires above 500 ha burned, regardless of ignition source.
- Burned area from large fires (B500): overall affected area from fires above 500 ha, regardless of ignition source
- Number of natural fires (NL): number of fires triggered by lightning.
- Burned area from natural fires (BL): overall burned area from fires triggered by lightning.
- Number of human fires (NH): number of fires triggered by an anthropogenic source.
- Burned area from human fires (BH): overall burned area from fires triggered by an anthropogenic source.

**2.3 Methods**

The methodology consisted of three stages. First, we explored changes in the temporal evolution of fire features by means of the Pettitt test and change detection procedures on a regional scale. In a second phase, a trend detection analysis was conducted using the Mann-Kendall test and the Sen's slope at regional and NUTS3 level. The third stage used a PCA to assess potential relationships among trends in fire features at NUTS3 level.

Statistical analyses, plotting and mapping were carried out using the R statistical software (R Core Team, 2016), packages *changepoint* and *trend* for change point analysis, *kendall* and *zyp* for trend analysis, and *ggplot2* for plotting and mapping the final results.

**2.3.1 Change point detection**

Change detection or change point detection aimed to identify times when the probability distribution of a time series changes. In order to identify change points in our time series we used four different tests.

First of all, the Pettitt test, a non-parametric method commonly applied to detect a single change-point in hydrological or climate series with continuous data (Pettitt, 1979). It tests the $H_0$: no change, against the alternative ($H_1$): a change point exists. One of the advantages of this technique is its robustness to deal with outliers. In the context of wildfire science, the Pettitt test has previously been applied to detect fire regime shifts as a consequence of policy and socio-economic development, in Pezzatti et al. (2013) and Moreno et al. (2014).

The Pettitt test is calculated using the following equation:

$$Ut, T = \sum_{i=1}^{t} \sum_{j=t+1}^{T} sgn\,(Xi - Xj) \tag{1}$$

Where $sgn(X) = 1$ for $X > 0$, $0$ for $X = 0$ and $-1$ for $X < 0$, and $T$ is the length of the time series in years. The probability of a significant change existing is calculated as follows:

$$p(t) = 1 - exp\left(\frac{-6 \cdot U_{t,T}^2}{T^3 + T^2}\right) \tag{2}$$





Where $|U_{t,T}|$ reaches the maximum value where the most significant change point is found (Pettitt, 1979). This methodology allows for the identification of the most probable change point in the period examined, in each fire feature by region and season. A specific function has been developed in R environment to calculate the change point using the Pettitt approach.

185    As an alternative method to the Pettitt test, three additional algorithms were applied; more specifically, the *cpt.meanvar* method to identify changes in mean and variance, calculating the optimal positioning of a change point for the input data (Chen and Gupta, 2000):

- AMOC (At Most One Change) method is the simplest expression of the change detection algorithms from the *changepoint* package. It can detect a single change point (Hinkley, 1970), much the same as the Pettitt test.

- PELT (Pruned Exact Linear Time) is one of the most widely used methods for change point detection. It can detect multiple change points in large data sets (Killick et al., 2012), unlike the Pettitt test or AMOC. It includes an enhanced optimal partitioning, leading to a substantially more accurate segmentation. This ensures minimum change point detection in a time series, regardless of the applied penalty value. Thus, PELT is known as a more precise algorithm, usually outperforming other methods such as Binary Segmentation. The CROPS (Changepoints for a Range of PenaltieS) penalising type was selected. The lower the pen.value, the higher the numbers of change points detected. For this reason, we chose many different minimum pen.values, in order to find at least one, or no more than two, breakpoints. One of the advantages of this last option is that continuous false change points were avoided commonly found at the beginning/end of the time series (for example, many cases with AMOC algorithm).

- BinSeg (Binary Segmentation) is an effective method for multiple change point detection (Scott and Knott, 1974). It searches for the first significant change point in a sequence, then breaks the original sequence into two sub-sequences: before and after the first significant change point. The procedure tests the two sub-sequences separately for a change point, with the process repeated until no further sub-sequences have change points (Chen and Wang, 2009). In our case, we previously defined a possible change point limited in 1 (Q=1), to obtain only the most significant. To this end, the default penalty parameter MBIC (Modified Bayes Information Criterion Penalty) was chosen, which has proved to reduce overestimation in the number of change points, and often detects the correct model (Bogdan et al., 2008). Therefore, there is no need to select a penalty value; hence in all the cases, this value is automatically established as 14.8.

### 2.3.2 Trend analysis

Change detection procedures determine if and when a certain feature has undergone a significant change across the study period. However, does it imply an increase or decrease in the values of that feature? Moreover, how strong is that change? Is the change stationary or does it vary over space? To answer all





these questions, we used a trend detection procedure combining the Mann-Kendall test (MK) and Sen's slope (SS).

MK is a non-parametric statistical test appropriate for identifying trends in time series of data (Kendall, 1975; Mann, 1945). It is suitable for detecting linear or non-linear trends (Hisdal et al., 2001; Wu et al., 2008). In this test, the null ($H_0$) and alternative hypotheses ($H_1$) are equal to the non-existence and existence of a trend in the time series of the observational data, respectively. Previous studies by San-Miguel-Ayanz et al. (2012) and Rodrigues et al. (2013) support the use of MK in the context of wildfire

science. MK main outputs are the *Tau* value and its associated significance level (*p*-value). *Tau* can be used to determine the sign of the trend, i.e. upward (*Tau*>0) or downward (*Tau*<0). Trends are considered significant when *p*-value<0.05. To facilitate the interpretation of MK outcomes, we calculated an aggregated parameter combining the *Tau* and *p-value*, the so-called *signed p-value*. It combines information on both sign and significance, calculated as the multiplication of the significance level either

by 1 when *Tau*>0, or by -1 when *Tau*<0.

The magnitude of the change was subsequently assessed by means of the Sen's slope (Sen, 1968), a nonparametric alternative for estimating the median slope joining all possible pairs of observations, which enables a comparison of the magnitude of the detected trends. Both MK and Sen's slope were calculated for all fire features by region and NUTS3 level, and for both seasons.


### 2.3.3 Principal Component Analysis and mapping

Finally, Principal Component Analysis (PCA) was carried out on Sen's slope's values in order to synthesize the detected changes. PCA is a widely used technique for summarizing a large set of variables into fewer and common factors, reducing the variance of the original data. Representative principal

components (PCs) were selected using the Kaiser Criterion (Kaiser, 1960), which only retains PCs with eigenvalues greater than 1. In turn, the Varimax Rotation (VR) method was applied to identify key trends. VR transforms the selected PC, maximizing the sum of the variance, obtaining higher coefficients or near to zero, thus, with fewer intermediate values. The objective is to link each variable to one maximum PC to make interpretation of PCA results easier (Horst, 1965; Kaiser, 1958).


Furthermore, we summarized the temporal behaviour retrieved from PCs on an additional map. Eigenvalues from PCs 1 and 2 were classified into four categories according to their sign (positive or negative trends) and significance level (above (significant) or below (non-significant)) a 90% Confidence Interval. PCs 3 and 4 were only shown when significant. In this way, we were able to outline

homogeneous areas according to the observed temporal evolution.






## 3   Results

### 3.1 Change Point detection

Figures 2 and 3 show the temporal evolution of fire features in the three pyroregions examined (NW, green line; HL, light blue line; MED, orange line) with their corresponding detected change points. Only those cases where at least three of the methods agree in the year of change are taken as change points. Table 1 summarizes the year(s) of change obtained by the four algorithms. The majority of change points were detected between the late 1980s and the early 1990s. Change points have been detected in the MED

region in N S, B, N500, B500, BL S, NH S and BH. Summer changes are consistently observed in 1994 in the MED features, whereas changes in the winter season in this region appear around in 1984-86. Additionally, another six change points were found in the NW region in NH W in 1987, NL W in 1993 and B S in 1991, and two in N W and B W, were found in 1984-87. Finally, one change point was detected in N S, but in 2005-2006. In HL, five change points were found in N W and NH W, both in

1987, also N500 S and BL W in 1990-1991, and finally BH S in 1977. It is important to note that some fire features in particular regions are very few, such as in HL and NW regions, which do not appear in Figures 2 and 3.

Table 1


Figure 2

Figure 3


### 3.2. Trends analysis

### 3.2.1 Region level

Table 2 summarizes the results from MK. Similar to change point analysis, the MED region stands out as the one with most significant changes. In general, the region returns mostly downward trends in all fire

features, significant in all cases and seasons, except NL, NH and BH during winter.  Only a few features underwent a trend either way in the NW region. Significant upward trends were detected in N, B, NH and BH. In all cases, trends occurred during the winter season. Significant downward trends were found in B and B500 during summer. Again, HL is the region with fewest significant trends. Overall, human-related features (NH and BH) show significant upward trends in summer and winter, whereas N increases during

winter.

Table 2



### 3.2.2 NUTS3 level

Trend detection analysis at NUTS3 level combines MK and SS. Maps in Figure 4 summarize the results of this section. Every single map displays the results for a given fire feature. The overall value of the feature, i.e. total number of fires, burnt area, number of human-caused fires, etc., is represented by symbol (circle) size. The colour of the circle indicates whether the MK test denotes a significant trend or not. Grey circles represent non-significant trends, whereas coloured symbols denote significant trends. For the significant trends, the value of the SS is plotted inside the circle. We used a green-yellow-red

colour ramp to represent both the sign of the trend (negative in green and positive in red), as well as trend magnitude according to the SS value.

Observing the spatial distribution of significant trends, an increment in N was found in provinces of the northwest area, and the western provinces of the hinterland on the border with Portugal. Similar to the results at regional level, provinces on the Mediterranean coast show a decrease in the number of fires,

although some provinces in the southern region (Andalusia) do not show significant trends. However, differences in the seasonal behaviour were observed, with increasing trends found in the eastern provinces in NW. In turn, provinces with significant decreasing trends were located along the Mediterranean coast, southern Andalusia and the majority of provinces in Galicia. Winter N clearly presents an increase for most of the NW region, and many areas of the HL.

This spatial distribution changes slightly for total B and summer B compared to those observed in N, being more visible in the northwest area (Galicia and Asturias), but also a large part of Andalusia, where negative trends play a decisive role at the expense of positive trends. On the other hand, NH shows more increases across the territory, except the Mediterranean coast. However, with BH, decreases are more evident, mainly concentrated in the north-east coastal areas, some provinces of the northwest and

Andalusia.

The spatial distribution of SS values across mainland Spain reveals a high spatial variability in trend magnitude. The strongest trends (SD <-1.64) or (SD >1.64) were found in the NW area, both positive and negative. Strong positive trends were identified in winter fire features relating to frequency, such as total N, and N - NH, but also with burned area, as with B and BH. On the other hand, the main negative trends

were located again in the NW region for some summer fire features like N, B, NH and BH. In addition, the rest of the territory is covered by intermediate values, mainly in total N and B, B S, N W and NH. However, in most areas moderate negative trends play a major role (especially in the Mediterranean coast) whereas moderate positive dynamics are concentrated in the western provinces of hinterland.

Figure 4

### 3.3 Principal Component Analysis and Varimax Rotation

PCA was applied to SS values. Results from PCA provide an overview of the most relevant links among trends in fire features. According to the Kaiser Criterion, 4 components (representing 88% of the



variance) out of the initial 14 were selected. Consequently, VR was only calculated for those 4 PCs. According to PCA eigenvalues (Table 3), PC1 (38% of the variance) is associated with changes in fire frequency, particularly the number of fires and human-caused fires during winter. PC1 gathers N (0.44), N W (0.52) and NH W (0.45). PC2 (27% of the variance) relates to the fire-affected area. B (0.47), B S (0.44), BH S (0.44) and N S (0.43), suggest that burned area trends are mainly related to summer human dynamics, and a slight increase in summer fire frequency. Large fires trends are noticeably isolated in PC3 (15% of the variance, both in terms of frequency and burned area). Finally, PC4 (8% of the variance) separates natural fires dynamics in the same way as large ones fall into PC3. In general, PC1 relates to winter fire frequency, PC2 to summer burnt area, PC3 to large fires and PC4 to natural fires.

Table 3

Figure 5 displays PC values at NUTS3 level. The NW and, to a lesser extent, MED regions show the highest magnitude of change when looking at the four PCs in the same picture. PC1 displays both the highest and lowest values in the NW region, although some provinces in the north-east area of the Mediterranean also show low values. PC2 shows higher values over the HL region and lower in the western area of NW. Lower values were observed for PC3 in some provinces of the Ebro Valley and others, such as Valencia, Cádiz and Ourense. On the other hand, some provinces in the western NW show moderately increased values (especially in Leon). However, PC4, which represents naturally-caused fires, exhibits intermediate positive values all over the study region, especially some provinces in the hinterland of the NW region, Valencia and Cádiz. Finally, the main negative values are located in several provinces on the Cantabrian (north) coast and the central Pyrenees (Huesca).

Figure 5

Figure 6 displays the summary of PCA. Increased fire frequency was observed only in the NW region, in the inland provinces of León, Zamora and Ourense, and also in the Cantabrian cornice. Nevertheless, the burnt area decreased throughout the region. A significant winter frequency decrease was solely found in Pontevedra. However, a non-significant winter frequency decrease was observed along the Mediterranean coast, and most of the interior of the country. In these latter areas, an increased summer burnt area was also observed. On the other hand, a significant decrease in the summer burnt area was only detected in the Galician provinces (NW). In addition, significant trends in large or natural fires were found in the 3 regions. Increased lightning fire activity was observed in Leon and Zamora (NW) and Valencia (MED). Lesser natural fire activity was detected in Asturias (NW) and Huesca (HL). In turn, the occurrence of large fires was more frequent in Leon and Pontevedra (NW), whereas the opposite could be found in Ourense (NW), Huesca (HL) and Valencia (MED).

Figure 6





## 4 Discussion

In this paper, we present an analysis of spatial-temporal trends of several fire regime features at different scales for mainland Spain. Various statistical methods for time series were applied to historical fire data to: (i) explore the temporal behaviour of fire features, and (ii) investigate key relationships in trends, with the end purpose of the research being to improve the definition of fire regime.

Change detection procedures suggest the existence of change points in several fire features (Table 1).
Changes were mostly found in the MED region from the late 1980s to the first half of the 1990s. Moreno et al. (2014) support our findings for N and B on a seasonal scale, and also found similar change points using the Pettitt test on a stepwise approach with an 11-year moving window. This work is a particularly good match for our objective, as the same regions and fire data from the EGIF database were used, although the study period was slightly different (1968-2010) and only examined N and B. In particular,
these authors observed downward changes starting from the 1990s to the present in the MED region for both W and S, and in the S of NW and HL, which are in line with our findings (see Figure 2). They concluded that climate might have played a role in the change points of the MED region (mid of 1980s and 1990s) and the NW region (1991). In addition, the change points we detected in the NW region for the N W feature (Figure 2) might be linked to different causes, such as rising population density,
agricultural activities, as well as more cases of arson, as Moreno et al. (2014) have pointed out. Overall, all methods detect significant changes in some fire features in the MED and the NW regions during both seasons, although slight differences in the reported year do exist.

We considered it necessary to assess other fire regime features, such as large fires and fire sources. The inclusion of trends in large fires is justified because of their remarkable socio-economic and natural
impact (Alvarado et al., 1998; San-Miguel-Ayanz et al., 2013). Change detection suggests that N500 has changed since the mid-1990s throughout the Mediterranean (see Figure 2), supported by findings from MK, which detects a decrease in frequency and affected area (Table 2). Cardil and Molina (2013) report similar changes, although these authors have taken large fires to be those burning more than 100 ha, and have excluded some provinces from their assessment. They and others, like Brotons et al. (2013) and
Ruffault and Mouillot (2015), suggest that large fires have decreased because extinguishing methods have improved.

Findings from change detection are supported and completed by those from trend analysis. At a regional level, the MED region shows a negative trend in the majority of fire features (see Table 2). NW and the HL share a positive trend during winter. N presents a general downward trend in both seasons in MED
and during summer in NW. In the case of the HL region, the trend in N suggests a higher frequency of fires in winter. This behaviour was also found by Zavala et al. (2011) and Turco et al. (2016). The latter found an apparent shift in the mid 1980's, the same as we detected. Among the feasible causes that may explain this spatial contrast, we found factors such as land use changes caused by land abandonment leading to vegetation recovery during recent years (Bonet and Pausas, 2007; Castellnou et al., 2010),
resulting in an accumulation of fuels. However, B shows a decrease in all regions and seasons. Previous studies by Rodrigues et al. (2013), Spano et al. (2014) and Turco et al. (2016) have also found negative trends for a very similar time span. These works suggest that the decrease might be explained by recent



improvements in management of wildfires and monitoring systems. Additionally, the European Forest Fire Information System (EFFIS) observed a clear downward trend in the average fire size in some
southern European countries (including Spain) after 1990, partly due to improved fire protection services (Schmuck et al., 2010). Nevertheless, the NW region during winter emerges as the sole exception, with the affected area showing an upward trend. It can be argued that a persistent increase in arson has tended to extend the surface affected in this area (Ganteaume et al., 2013). Finally, positive trends are detected in winter in NH in the NW region, and in the HL region (both S and W), contrasting with a decrease found
in the MED region. BH follows a positive trend mainly during winter in the NW and in many areas in the south, whereas during summer a decrease is more widespread. This fact suggests that summer BH is declining, but intensifies during winter. The reasons which may explain this fact could relate to a continuance of arson attacks as a common practice which still remains. To our knowledge, this is the first analysis of fire frequency and burned area based on the source of the fire. Therefore, we cannot establish
any comparison.

Downscaling to NUTS3/province level, SS has exposed the underlying spatial heterogeneity in the magnitude of the trends, either positive or negative (Figure 4). In this respect, N is the feature with the highest degree of change. Trend magnitude in N appears to be distributed along a west-to-east gradient, starting with increasing dynamics in the west, and ending with downward trends in the eastern
Mediterranean provinces. On the other hand, the sharpest decreases are observed in features relating to burned areas such as B, B during summer, also BH during summer. However, this behaviour is reversed during winter, although trend magnitude is less marked. This is probably due to the improvement in fire extinguishing or encouraging monitoring and prevention (MAPA, 1988; Rodrigues et al., 2016), particularly encouraged during summer. Fires ignited by lightning perhaps show the most contrast as
there is a marked dichotomy between west and northeast. For instance, decreasing dynamics are found in the northeast area, whereas major upward trends are situated in the western half of the region, which is considered one of the most lightning-ignition prone areas of Spain (Ortega et al., 2012). According to our results, there seems to be increased fire activity from natural causes. However, trends are more noticeable in NL than in BL, thus the average size of lightning-caused fires seems to be shrinking.

Finally, PCA-VR enables trends to be grouped to provide an easily readable description and characterisation of fire regime at provincial level, also clarifying the spatial pattern of key fire trends. Therefore, we extracted four main components (see Table 3, Figure 5 and 6), i.e. four distinct temporal behaviours: winter fire frequency (PC1), summer burnt area (PC2), large fires (PC3) and natural fires (PC4). The first two components are associated with seasonal fire activity, whereas components 3 and 4
relate to intrinsic characteristics of wildfires, such as fire size and ignition source, respectively. PC1 has led to identifying winter frequency trends (N W) and human-caused frequency during winter (NH W) as key trend features, while PC2 gathers trends from burned area features, but mostly summer trends (B S and BH S), indicating that summer fire dynamics might play a secondary role compared to winter, at least in terms of the strongest temporal trend. On the other hand, the last two PCs, large fires and natural fires
trends, appear to be similarly important. In addition, a seasonal contrast is clearly evident between the NW region dynamics and the rest of mainland Spain (i.e. negative trends located mainly in Galicia).



**5 Conclusions and future work**

In this paper, we have explored spatial-temporal changes of several fire regime features in Spain at regional and provincial levels. To this end, we combined change point detection techniques, trend detection procedures and PCA, applied to fire data from 1974-2013. Our results suggest that two main trends based on seasonal differences can be distinguished: fire frequency during winter and burned area during summer. It is important to highlight that in both cases, human cause is strongly correlated to the trends, and thus apparently changes in burned area and fire frequency are partially controlled by human-caused fires. Additionally, mapping SS and PCA results at NUTS3 level suggests different behaviour in the northwest provinces, which return the highest values, both in terms of frequency (upward trends) and burnt area (downward trends).

Change detection suggests a main breakpoint in the temporal evolution of fire features around the late 1980s and in the first half of the 1990s. On the other hand, the MK test on a region scale has revealed that the MED region presents a high degree of negative trends in the majority of fire features, opposed to the Hinterlands and Northwest. According to SS, the main trends at NUTS3 level show high spatial-seasonal variability, and several trend gradients linked to the number of fires and naturally-caused fires were detected. In this regard, overall fire frequency shows an upward tendency, particularly strong during winter while the burned area exhibited a general downward trend.

The analysis of spatial-temporal trends opens new research lines. Firstly, further evaluation is required to incorporate other benchmark spatial units to provide greater detail than found at provincial level (for instance, grid-cells). Nevertheless, deeper insights into causes explaining temporal behaviour of the main fire regime features should be explored, especially those linked to weather conditions and land use changes. Finally, the role of small fires (1 < ha) can be included, thus enriching fire regime assessment in order to avoid potential bias caused by their exclusion. In any case, the analysis given in this paper should provide a useful reference to obtain spatially and temporally explicit assessment of fire regime changes, to help improve delimitation of pyroregions and to gain a more complete overview of wildfire phenomenon.

**Acknowledgments** The Spanish Ministry of Education has financed this work: FPU grant 13/06618.

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





**Table 1: Change points for AMOC, BinSeg, PELT methods and Pettitt test (\*significant changes p-value <0.05) by fire feature, region and season from the period 1974-2013. Grey shaded features indicate matching probable changes in at least three methods.**

| | NW | | | | HL | | | | MED | | | |
|---|---|---|---|---|---|---|---|---|---|---|---|---|
| | Summer | | | | Summer | | | | Summer | | | |
| | AMOC | BinSeg | PELT | Pettitt | AMOC | BinSeg | PELT | Pettitt | AMOC | BinSeg | PELT | Pettitt |
| N | - | 2006 | 2000; 2006 | 2005 | - | 1977 | 1977 | 1983 | 1994 | 1994 | 1994 | 1994* |
| B | - | 1990 | 1990 | 1991* | - | 1977 | 1977 | 1991 | - | 1994 | 1994 | 1994* |
| N500 | - | 2006 | 2006; 2008 | 1990 | - | 1991 | 1981; 1983; 1991 | 1991 | 1994 | 1994 | 1995; 1997 | 1994* |
| B500 | - | 2006 | 2006; 2008; 2010 | 1991 | - | 1977 | 1977 | 1991 | - | 1994 | 1992; 1994 | 1994* |
| NL | - | 2006 | 1982; 1984 | 1988* | - | 2006 | 1980; 2006 | 1980 | 2011 | 1994 | 2011 | 1996 |
| BL | - | 2006 | 2006; 2010 | 1988 | - | 1990 | 1990 | 1995 | 1995 | 1994 | 1994 | 1994 |
| NH | - | 1976 | 2000; 2006 | 2006 | - | 1977 | 1977 | 1997* | - | 2005 | 1991; 1993 | 1994* |
| BH | - | 2008 | 2008 | 1990 | - | 1977 | 1977 | 1977 | - | 1994 | 1990; 1994 | 1994* |
| | Winter | | | | Winter | | | | Winter | | | |
| | AMOC | BinSeg | PELT | Pettitt | AMOC | BinSeg | PELT | Pettitt | AMOC | BinSeg | PELT | Pettitt |
| N | - | 1984 | 1984 | 1987* | - | 1987 | 1987 | 1987* | - | 2005 | 1995, 1997 | 1999 |
| B | - | 1984 | 1984; 1989 | 1987* | - | 1989 | 1989 | 2000 | 1985 | 1985 | 1985 | 1986* |
| N500 | - | 1989 | 1990; 1993; 1995 | 1996 | - | 1977 | 1995; 2009 | 1989 | 1984 | 1984 | 1992; 1999 | 1986* |
| B500 | - | 1984 | 1990; 1993 | 1996 | - | 1977 | 1995; 2009 | 1989 | 1984 | 1984 | 1992; 1999 | 1986* |
| NL | 1994 | 1993 | 2000; 2007 | 1993 | - | 1976 | 1976 | 1990 | - | 1977 | 1977 | 1992 |
| BL | - | 1984 | 2002; 2007 | 1993 | 1990 | 1989 | 1990; 1994 | 1990 | - | 1977 | 1977 | 1999 |
| NH | 1987 | 1987 | 1987 | 1987* | - | 1987 | 1987 | 1987* | - | 2005 | 2005 | 2005 |
| BH | - | 1984 | 1984 | 1987* | - | 1989 | 1989 | 1987 | 1981 | 1981 | 1981 | 1986* |

**Table 2:** *Signed p*-value of MK test in the period 1974-2013 by fire feature, season and region. Values in bold
and highlighted in grey correspond to significant trends (*p-value*<0.05), with their corresponding symbol + or –
for positive or negative trends, respectively.

| | N | | N500 | | NL | | NH | |
|---|---|---|---|---|---|---|---|---|
| | S | W | S | W | S | W | S | W |
| NW | - (0.14) | + (0.01) | - (0.03) | + (0.26) | + (0.14) | - (0.49) | + (0.77) | + (0.01) |
| HL | + (0.21) | + (0.01) | - (0.22) | - (0.39) | + (0.62) | - (0.03) | + (0.01) | + (0.01) |
| MED | - (0.01) | - (0.01) | - (0.01) | - (0.01) | - (0.11) | + (0.13) | - (0.01) | - (0.54) |

| | B | | B500 | | BL | | BH | |
|---|---|---|---|---|---|---|---|---|
| | S | W | S | W | S | W | S | W |
| NW | - (0.01) | + (0.03) | - (0.13) | + (0.41) | + (0.68) | - (0.28) | - (0.13) | + (0.01) |
| HL | - (0.25) | - (0.95) | + (0.51) | - (0.48) | - (0.44) | - (0.08) | + (0.84) | + (0.46) |
| MED | - (0.01) | - (0.01) | - (0.01) | - (0.01) | - (0.22) | + (0.06) | - (0.01) | + (0.01) |





**Table 3: VR correlation values, standard deviation (SD) and variance (VAR) from PCA on SS results, 1974-2013. The most meaningful features (correlation |>0.43| or |<-0.43|) are highlighted in light grey.**

| | | PC1 | PC2 | PC3 | PC4 |
|---|---|---|---|---|---|
| | SD | 2.3 | 1.95 | 1.47 | 1.07 |
| | VAR | 0.38 | 0.27 | 0.15 | 0.08 |
| **Fire feature** | N | 0.44 | 0.23 | | -0.14 |
| | N S | 0.18 | 0.43 | | -0.11 |
| | N W | 0.52 | | 0.11 | -0.10 |
| | B | | 0.47 | | |
| | B S | -0.18 | 0.44 | 0.11 | |
| | B W | 0.35 | | | 0.21 |
| | N500 S | | | 0.63 | |
| | B500 S | | | 0.69 | |
| | NL S | | | | 0.60 |
| | BL S | | | | 0.60 |
| | NH S | 0.19 | 0.36 | -0.17 | 0.12 |
| | NH W | 0.45 | | | |
| | BH S | -0.20 | 0.44 | -0.14 | 0.27 |
| | BH W | 0.27 | | -0.14 | 0.29 |









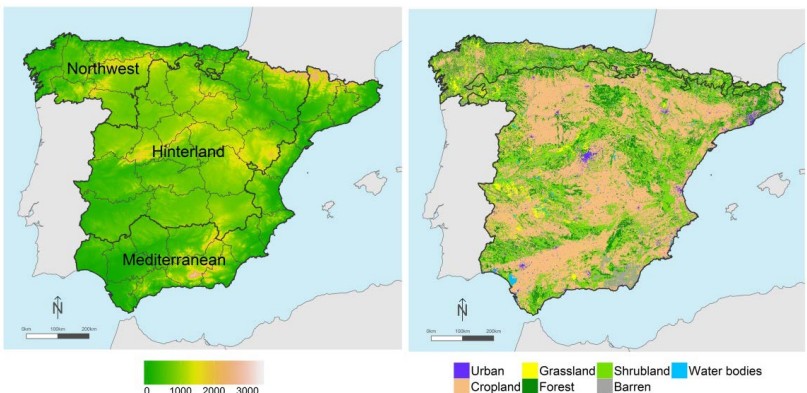

**Figure 1**: Spatial distribution of pyroregions and NUTS3 units in Spain with an elevation MDE (m) (left) and
biogeographical limits with generalized land cover from CLC 2006 (right).









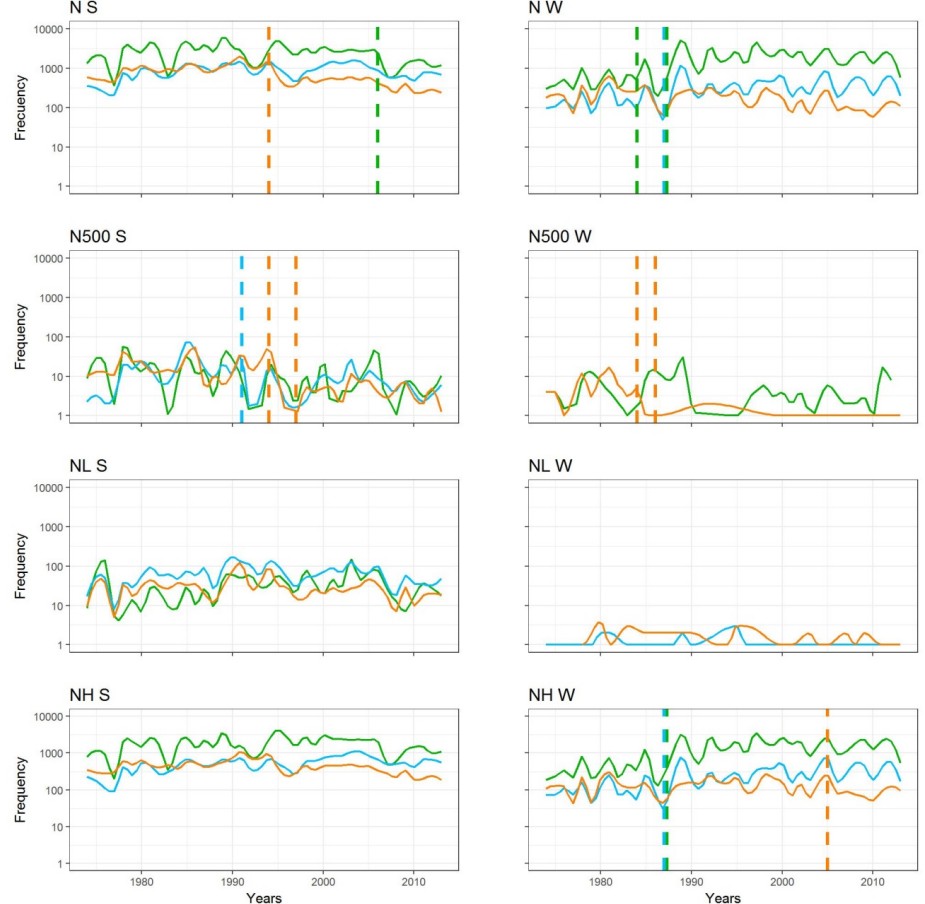

**Figure 2: Change points and temporal evolution of fire frequency features (log scale) in the period 1974-2013 at region level: NW green line, HL light blue, and the MED orange line. The column on the left refers to the summer season, winter is on the right. Dashed vertical lines represent probable change points.**








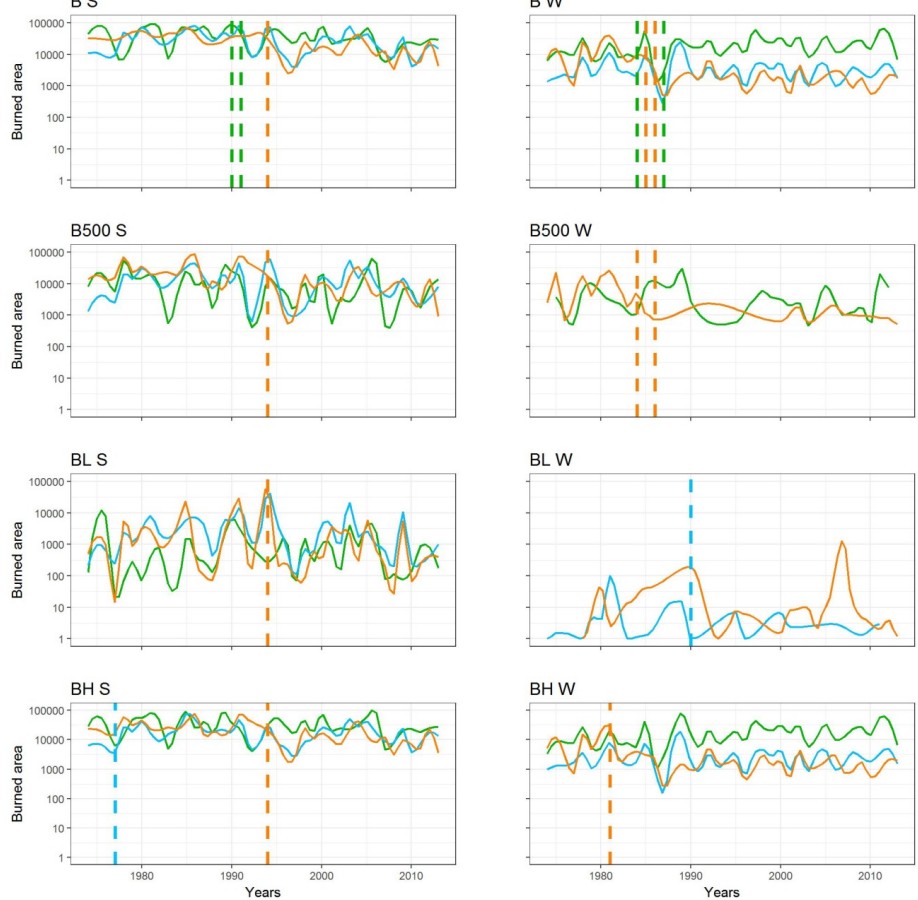

**Figure 3: Change points and temporal evolution of burned area features (log scale) in the period 1974-2013 at**
**region level: NW green line, HL light blue, and the MED orange line. The column on the left refers to the**
**summer season, winter is on the right. Dashed vertical lines represent probable change points.**






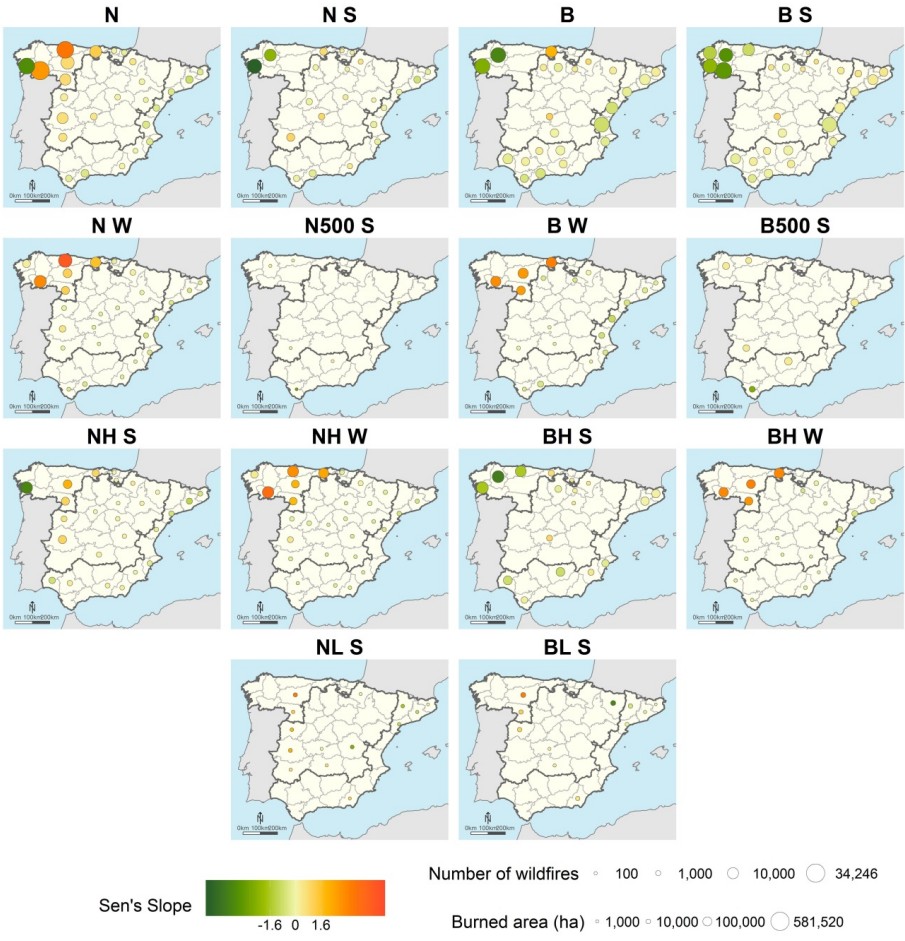


**Figure 4: Spatial distribution of significance level of SS values 1974-2013. Proportional symbols represent the number of wildfires and burned area value. SS value displayed in colour using standard deviation intervals. Provinces without symbols represent non-significant trends according MK.**





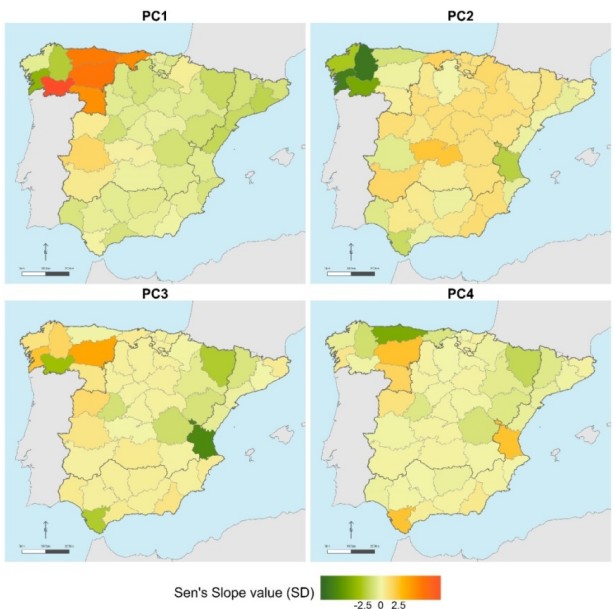

**Figure 5: Spatial distribution of the principal component coefficients of SS, 1974-2013. Values represented using standard deviation intervals.**

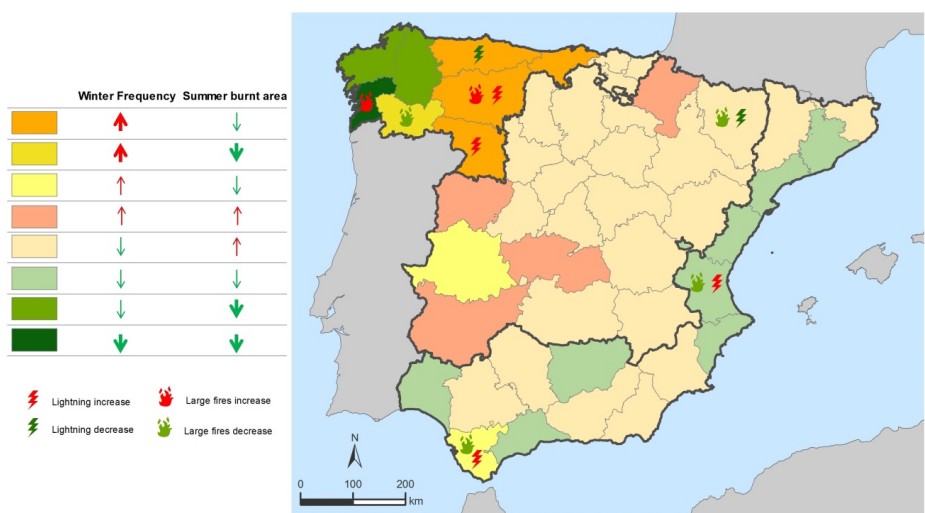

**Figure 6: Summary of spatial-temporal behaviour from PCA. Bold arrows mean significant trends, and thin ones non-significant trends according to 90% CI.**