# Peer review of "Exploring spatial-temporal dynamics of fire regime features at mainland Spain"

_Natural Hazards and Earth System Sciences, 2017_

## Referee Comment (RC1) · Anonymous Referee #1 · 22 Jul 2017

In general, this is a clear, objective, well organized and written manuscript. My major concern is about the novelty of this study. Apparently, this topic is not new in Spain. The authors cited a significant number of studies on this theme. The motivation is obviously associated the well-known changes/trends in the fire incidence in Spain. It is even cited a study performed for the same regions and the same (EGIF) fire dataset although the study period is slightly different (1968-2010). The authors justify the paper with the inclusion of other fire regime features but I'm convinced if it is sufficient. Besides this problem, I only have a few number of general and a small number of specific questions, comments and/or suggestions. This study aims to detect break/change point and trend in time series of several fire regime features in Spain on different spatial scales/basis. The abstract is a good summary of the paper and the introduction comprehensively

cover all important aspects to the study.

My general comments/suggestions/questions on the other sections of the manuscript and are the following: Study area Figure 1 do not helps to understand the location and size of the Eurosiberian and Mediterranean regions. The description of the type of climate is not very accurate, in the sense that any figure is presented or any study is cited. The AEMET/IM Iberian Climate Atlas, Peel et al. (2007), Kottek et al. (2006), among other could be cited and used. The three considered pyroregions present some similarities but also some differences to other studies not cited. For example, Sousa et al. (2015) and Trigo et al. (2016) identified different pyroregions. These similarities/differences should be discussed because they could have significant impacts on the obtained results.

Data The quality of the datasets is one of the most important aspects on this type of studies. The authors identified some completeness problems for small fires (burnt area < 1ha). The same type of problem was found for Portugal (Please see Pereira et al., 2011) whereby this study could be cited. Besides this aspect, what type of data quality analysis was performed on the fire dataset? Another important aspect is the size of the dataset. It is very important to know the size (number of fires) of the dataset as well as how many fires are in each group (NH, NS, N500, N500N N500S, NL, NH, etc.) as well as on each province/NUTS3 region. Please provide this information on the manuscript. Finally, since the authors do not provide the intra-anual distribution of any fire regime feature, it not possible to understand the splitting of the annual data in to the summer (April-September) and winter (October-March). In fact, according to Sousa et al. (2015) and Trigo et al. (2016), it would make more sense another split (May-November and December-April). The authors should validate their options and discuss these aspects in the manuscript.

Methods The authors describes the characteristics of the used methods. However, it is also important to explain which other methods could have been applied for the same purpose and why these methods were selected. It is also important to explain why you

limit the number of detect breaks to just 1.

Discussion This section need to be improved; sometimes, is just a repetition of the results presentation; others cases, studies with similar findings are cited; this is not the best/proper validation/interpretation of the results. For example, in line 38, the decreasing trend in MED region is justified with the study of Moreno et al. (2014) which suggested that "climate might have played a role in the change points". However, the questions is the following: did Moreno or the authors detect any change in the climate? Even if those change occurred what is the impact on the fire regime features? The same happens, for example, in lines 384-385 and lines 395-396. In this later case, this means more or better/more efficient methods? How this improvement was assessed?

Reference list can be updated/enlarged.

Specific questions /comments/suggestions Line 260-261, should not be in the main text but part of the figure caption; Line 294, the caption of figure 4 is not clear; at the first reading only mention SS; Line 317, SD is not defined; Lines 409-410, average fire size is a very "dangerous" measure, especially due to data errors. This is recognized by the authors when removed small fires (burnt area <1 ha) from the analysis.

Tables & Figures

Tables and figure should be self-explanatory. Therefore, for example, explain/describe all acronyms, symbols, etc.

Table 3. Please explain how the thresholds (-0.43 and 0.43) were obtained to define "The most meaningful features".

Figure 1. It is not clear if the named regions are the pyroregions; the "continuous" color scale is not a good option; it is virtually impossible for the common human eye to identify the associated value. This is also valid for figure 4 and figure 5. The presented CLC nomenclature is not the usual/official one. Please explain how was defined, i.e., which CLC classes are urban (eventually all the Artificial classes), grassland, shrubland, etc.

The climate classification can be presented, for example, in panel b.

Figure 4. Caption is contradictory; first mention "Spatial distribution of significance level of SS values 1974-2013" and, in the end, "Provinces without symbols represent non-significant trends according MK".

Figure 6. A "Table" and a Figure do not seems a good idea. What don't you plot two figure, one for summer and other for winter and, in each case you only plot the "statistically significant" arrows?

AEMET, I. (2011). Iberian climate atlas. Agencia Estatal de Meteorología (España) and Instituto de Meteorología (Portugal), Madrid, Spain.

Kottek, M., Grieser, J., Beck, C., Rudolf, B., & Rubel, F. (2006). World map of the Köppen-Geiger climate classification updated. Meteorologische Zeitschrift, 15(3), 259-263.

Peel, M. C., Finlayson, B. L., & McMahon, T. A. (2007). Updated world map of the Köppen-Geiger climate classification. Hydrology and earth system sciences discussions, 4(2), 439-473.

Pereira, M. G., Malamud, B. D., Trigo, R. M., & Alves, P. I. (2011). The history and characteristics of the 1980-2005 Portuguese rural fire database. Natural Hazards and Earth System Sciences, 11(12), 3343.

Sousa, P. M., Trigo, R. M., Pereira, M. G., Bedia, J., & Gutiérrez, J. M. (2015). Different approaches to model future burnt area in the Iberian Peninsula. Agricultural and Forest Meteorology, 202, 11-25.

Trigo, R. M., Sousa, P. M., Pereira, M. G., Rasilla, D., & Gouveia, C. M. (2016). Modelling wildfire activity in Iberia with different atmospheric circulation weather types. International Journal of Climatology, 36(7), 2761-2778.

2017-173, 2017.

---

## Referee Comment (RC2) · Anonymous Referee #2 · 31 Jul 2017

This temporal analysis of fire regimes features in Spain may be a very valuable addition to the fire science field, as it considers traits of fire regime characterization not contemplated before, beyond the usual number of fires and burned area, from a temporal perspective. There are many previous studies on how climate, topography, vegetation, and land use influence fire regimes, characterized by number of fires or fire frequency, severity/intensity, size of burned area or pattern. As there is abundant previous work on fire regimes characterization, the factor that set this analysis aside and merits publication is the application of change and trend detection procedures to fire features of special interest in Spain (i.e. large fires over 500 ha), and the PCA-Varimax Rotation applied to summarize trends. Procedures, though, may be applied elsewhere at different spatial and temporal scales.

[Figure]

However, the authors state that their temporal analysis aims "to refine and improve the spatial outline of fire regimes" and has an "ultimate goal of characterizing fire regimes". How it is proposed that their temporal variability in fire regime features is considered when defining fire regimes ? (line 50). It is unclear how they propose this to be done, or how their stratifications in space (three regions, provinces NUT3 level) and time (two fire seasons in winter-spring and summer, line 140) correspond to fire regime stratifications in Spain by other authors like Moreno & Chuvieco 2012 (four regimes), or official Spanish reports (that need citation (line 120). Other partitions of the territory were possible, and these pyroregions need better justification and definition. Some descriptive statistics of the fire database in the 2.2 Fire data section would probably help to justify the spatial and temporal stratification used.

Methods are well described and applied, though the use of fires over 1ha eliminates from the analysis a very large number of fires that would largely influence results related to number of fires features in the paper. However, this has been done also in other work i.e. Moreno & Chuvieco 2012 based on lack of accuracy of these data in older reports in the study period. Lines after 185 explain three algorithms for change point detection. Why settings were determined to find at least one, but no more than two breakpoints in PELT, and one (Q=1) in BinSeg? This makes sense for comparison purposes with AMOC and Pettitt, but is there not a risk to miss other significant changes?

There are some other minor issues that should be considered:

-The authors refer to CCAA in Spain the international readers will not be familiar with, i.e. Andalusia, Galicia or Asturias, not in Figure 1. Labels seem to be missing. What is the black line crossing the land cover map?. Regarding Figures 4 and 5, Sen's slope values are hard to distinguish.

-Why is the level for correlation in table 3 set to 0.43? Please explain.

2017-173, 2017.

---

## Author Comment (AC1) · 2 Aug 2017

REVIEWER 1: My major concern is about the novelty of this study. Apparently, this topic is not new in Spain.

AUTHORS: First of all, we would like to thank the reviewer for his/her comments and suggestions about the manuscript. We have tried to amend and address all pointed issues.

Regarding the novelty of our work, indeed this topic is not new in the context of Spain and we were very aware of that. However, we believe that our work goes a step further in providing insights and analyse dynamics in fire regime features. Specifically we have (i) extended the analysis to other fire regime features in our change-point analysis; (ii)

we also apply traditional trend analysis to these other features; (iii) assessing not only the sign of trends, but its magnitude, which has not yet been addressed; (iv) at different scales; (v) this work would also allow progress in the fire regimes zoning; (vi) finally, our most novel contribution is exploring the relationships and association among trends in fire features using Principal Components in an effort to provide a more synthetic interpretation as well. We also provide a summary map of the main trends detected which allows to outline homogeneous zones of temporal dynamics at province level. To our knowledge such kind of analysis and cartographic outputs has not yet been done.

REVIEWER 1:(Study area) Figure 1 do not helps to understand the location and size of the Eurosiberian and Mediterranean regions.

AUTHORS: We have removed the bioregions layer in the second map of Fig. 1, because we finally considered that this information is not truly necessary for the study area description.

REVIEWER 1: The description of the type of climate is not very accurate, in the sense that any figure is presented or any study is cited. The AEMET/IM Iberian Climate Atlas, Peel et al. (2007), Kottek et al. (2006), among other could be cited and used.

AUTHORS: We have improved the climate types' description and added the AEMET/IM Iberian Climate Atlas reference.

REVIEWER 1: The three considered pyroregions present some similarities but also some differences to other studies not cited. For example, Sousa et al. (2015) and Trigo et al. (2016) identified different pyroregions. These similarities/differences should be discussed because they could have significant impacts on the obtained results.

AUTHORS: We added these two references and briefly discussed the similarities/differences between their zoning and our pyroregions. In any case, we are using the 'official' regions provided by Spanish authorities because they are fully adapted to the way in which fire events are reported and spatialized.

REVIEWER 1:(Fire data) - The quality of the datasets is one of the most important aspects on this type of studies. The authors identified some completeness problems for small fires (burnt area < 1ha). The same type of problem was found for Portugal (Please see Pereira et al., 2011) whereby this study could be cited. Besides this aspect, what type of data quality analysis was performed on the fire dataset?

AUTHORS: We have included this reference in Fire data section (Pereira et al. 2011). Regarding to the data quality analysis, fire data comes from EGIF database, which generally guarantees high quality. However, we have found several grid coded errors that were corrected, and also discarded few records whose location was reported outside Spain limits. Apart from that, no other quality analysis has been done.

REVIEWER 1: Another important aspect is the size of the dataset. It is very important to know the size (number of fires) of the dataset as well as how many fires are in each group (NH, NS, N500, N500N N500S, NL, NH, etc.) as well as on each province/NUTS3 region. Please provide this information on the manuscript.

AUTHORS: This is a very good suggestion. Indeed we should have already provided this. We have added a table with this information.

REVIEWER 1: Finally, since the authors do not provide the intra-annual distribution of any fire regime feature, it not possible to understand the splitting of the annual data in to the summer (April-September) and winter (October-March). In fact, according to Sousa et al. (2015) and Trigo et al. (2016), it would make more sense another split (May-November and December-April). The authors should validate their options and discuss these aspects in the manuscript.

AUTHORS: We appreciate your suggestion; however, we have done the seasonal split mostly according to the fire danger seasons established by the Autonomous Communities legislations. Moreover, the seasonal partition proposed by others authors does not match the intra-annual distribution of some fire features such as natural fires.

We have run some tests exploring the May-November and December-April seasons and the results do not change significantly so they are not sensitive to those differences in the cut-off months. On the other hand, we plan to incorporate climate information in further developments and applications of our proposal (for instance. for fire regime zoning) in which we believe our seasonal split fits best.

We have included part of this justification in the corresponding section and the different tests addressed in the Discussion.

REVIEWER 1:(Methods) - The authors describe the characteristics of the used methods. However, it is also important to explain which other methods could have been applied for the same purpose and why these methods were selected. It is also important to explain why you limit the number of detect breaks to just 1.

AUTHORS: We have selected the most commonly employed methods in the literature and specifically to fire data for trend analysis. However, precisely because we weren't sure of the performance of Pettit and didn't find any work comparing Pettit to other methods, we explored other possibilities for the change point analysis, to determine if there is any variation depending on the method and also be able to report a 'consensus' result rather than a single one. We didn't limit change point detection to 1. It is true that Pettit and AMOC methods only are able to detect 1 point, but PELT reports more than one. The thing is that most of the time there is only 1 point detected, although in those cases where more than one was detected we reported them (Table 1).

REVIEWER 1:(Discussion)- This section need to be improved; sometimes, is just a repetition of the results presentation; others cases, studies with similar findings are cited; this is not the best/proper validation/interpretation of the results. For example, in line 38, the decreasing trend in MED region is justified with the study of Moreno et al. (2014) which suggested that "climate might have played a role in the change points". However, the questions are the following: did Moreno or the authors detect any change in the climate? Even if those change occurred what is the impact on the fire regime

features?

AUTHORS: The reviewer raises an interesting concern. Regarding to line 38, the work by Moreno et al.2014 detects climate influence in upward changes in all fire regimes, regions and vegetative season. Also in some downward change points of the Mediterranean and Northwest. However, the authors explicitly do not find or mention a real climate change beyond its influence on fire metrics.

Pausas and Keeley, 2009 review the importance of fire has waxed and waned in association with changes in climate and paleo atmospheric conditions.

Pausas 2004 reports a significantly relation between burned area variability and summer rainfall.

Pausas and Fernández-Muñoz concluded that the fire regimen change in Valencia cannot be explained by gradual climate change observed.

Turco et al. 2014 assessed the impact of climate changes incorporating regional climate models, which captures quite well the observed trends. However, they admitted the complex impact of climate change in burnt area, because of the triple relationship (climate-fuel-fire). They estimated an increase in fire frequency and a stable o slight decrease in burnt area in hotter scenarios.

Moriondo et al. 2006 found an increase in fire risk in two future scenarios for the entire Mediterranean region. Specially, fire features such as increase in number of seasons with fire risk, increase in the number and length of extreme events contribute in a great extent.

Salis et al. 2014 did no address the climate influence in wildfire regime, but the weather relation.

Venäläinen et al. 2014 concluded that weather and climate are the major factor controlling fires, but not the only ones. In addition, fires were only related to current-year climate variables.

REVIEWER 1: The same happens, for example, in lines 384-385 and lines 395-396. In this later case, this means more or better/more efficient methods? How this improvement was assessed?

AUTHORS: Regarding to lines 384-385 we share with Moreno et al. the fact that in the NW region human factors play an important role in terms of fire activity and fire trends. On the other hand, in lines 395-396, the first reference refers only by the introduction of new fire policy (fire suppression and prevention practises). The improved was assessed by means of a statistical framework based on spatially explicit daily fire occurrence data, the corresponding weather variables and the associated fuel moisture derived from a process-based model. The second reference investigated the role of fire suppression strategies in synergy with climate change on the resulting fire regimes in Catalonia, Spain. They addressed this issue with a spatially-explicit fire-succession model at the landscape level to test if the use of different firefighting opportunities related to observed reductions in fire spread and sizes.

REVIEWER 1:(Specific comments) - Line 260-261, should not be in the main text but part of the figure caption.

AUTHORS: We have moved this line to both figure captions (Figures 2 and 3)

REVIEWER 1: Line 294, the caption of figure 4 is not clear; at the first reading only mention SS.

AUTHORS: We have rephrased the sentence of this line to be more explanatory.

REVIEWER 1: Line 317, SD is not defined.

AUTHORS: We have defined SD as "standard deviation".

REVIEWER 1: Lines 409-410, average fire size is a very "dangerous" measure, especially due to data errors. This is recognized by the authors when removed small fires (burnt area < 1 ha) from the analysis.

AUTHORS: We have replaced the "average fire size" by "total burnt area" in this line.

REVIEWER 1:(Tables & Figures) - Tables and figure should be self-explanatory. Therefore, for example, explain/describe all acronyms, symbols, etc.

AUTHORS: We have incorporated the full name of all acronyms in the captions, or a section reference regarding to fire features description.

REVIEWER 1: Table 3. Please explain how the thresholds (-0.43 and 0.43) were obtained to define "The most meaningful features".

AUTHORS: This threshold was stablished based on the actual values we retrieved from PCA-Varimax. There is no rule-of-thumb when it comes to determine a correlation threshold. We now realise that reporting a cut-off value of 0.43 it's rather awkward. In fact, the actual value is 0.4 but, again, this is based on the two most correlated featured in each component.

REVIEWER 1: Figure 1. It is not clear if the named regions are the pyroregions; the "continuous" color scale is not a good option; it is virtually impossible for the common human eye to identify the associated value. This is also valid for figure 4 and figure 5.

AUTHORS: In Figure 1, we have included in the caption a pyroregions description while we have removed the elevation colour variable. In Figures 4 and 5 we have changed the continuous colour scale to a discrete colour scale for the variables mapped.

REVIEWER 1: The presented CLC nomenclature is not the usual/official one. Please explain how was defined, i.e., which CLC classes are urban (eventually all the Artificial classes), grassland, shrubland, etc.

AUTHORS: The CLC is a generalization or summary of all the land cover categories. We added how we defined them and which specific sub-category is within each one. The regrouping was as follows. Urban: all the artificial surfaces; Grassland: only pastures and natural grasslands; Shrubland: only moors and heathland, sclerophyllous vegetation and transitional woodland-shrub; Water bodies: all wetlands and water

bodies; Cropland: all agricultural areas (except pastures); Forest: only broad-leaved, coniferous and mixed forest; Barren: all open spaces with little or no vegetation.

REVIEWER 1:Figure 4. Caption is contradictory; first mention "Spatial distribution of significance level of SS values 1974-2013" and, in the end, "Provinces without symbols represent non-significant trends according MK".

AUTHORS: We appreciate this observation, we mean that we have finally selected the significance Sen's slope values according to the Mann-Kendall test, because the Sen's Slope doesn't report significance. Thus, we discard the provinces with non-significant trend according this last test. We have rephrased the sentence so as not to be confused.

REVIEWER 1: Figure 6. A "Table" and a Figure do not seems a good idea. What don't you plot two figure, one for summer and other for winter and, in each case you only plot the "statistically significant" arrows?

AUTHORS: We appreciate your suggestion, but we believe that adding another map here can saturate this figure to the detriment of the effort to summarize the main trends. On the other hand, we have previously divided both seasons between the components 1 and 2 in Figure 5. Finally, it is important to note that the table which accompanies Figure 6 is actually its legend. We have explored and tried different versions of this figure and in the end this was the better way to show and summarise our findings.

---

## Author Comment (AC2) · 10 Aug 2017

REVIEWER 2: This temporal analysis of fire regimes features in Spain may be a very valuable addition to the fire science field, as it considers traits of fire regime characterization not contemplated before, beyond the usual number of fires and burned area, from a temporal perspective. There are many previous studies on how climate, topography, vegetation, and land use influence fire regimes, characterized by number of fires or fire frequency, severity/intensity, size of burned area or pattern. As there is abundant previous work on fire regimes characterization, the factor that set this analysis aside and merits publication is the application of change and trend detection procedures to fire features of special interest in Spain (i.e. large fires over 500 ha), and the PCA-Varimax Rotation applied to summarize trends. Procedures, though, may be applied

elsewhere at different spatial and temporal scales. However, the authors state that their temporal analysis aims "to refine and improve the spatial outline of fire regimes" and has an "ultimate goal of characterizing fire regimes". How it is proposed that their temporal variability in fire regime features is considered when defining fire regimes? (line 50). It is unclear how they propose this to be done, or how their stratifications in space (three regions, provinces NUT3 level) and time (two fire seasons in winter-spring and summer, line 140) correspond to fire regime stratifications in Spain by other authors like Moreno & Chuvieco 2012 (four regimes), or official Spanish reports (that need citation (line 120). Other partitions of the territory were possible, and these pyroregions need better justification and definition.

AUTHORS: First at all, we would like to thank the reviewer for his/her useful comments and suggestions about the manuscript. We really appreciate the positive evaluation about our work. Indeed, there are many works devoted to this subject, and thus it is not easy to bring some novelty. We are particularly grateful for appreciating the novelty of our proposal. Regarding to the application of the procedures at different spatial and temporal scales, we would like to bring some light here. In fact, we are currently working in a new fire regime zoning in which we are including trend magnitude as a key parameter because we believe that a complete fire regime characterization should account for at least the dynamics of the main fire features. This work provides enough evidence of changes in fire features; therefore, we can infer that fire regime zones may not be the same in 1974 than in 2013, something that is assumed in current works, for instance Moreno & Chuvieco 2012. Bringing this up here was not possible since we have limited space. However, the way in which we propose this to be done is, for example, by using trend outputs as another input of the cluster or zoning algorithm. This also would involve downscale the spatial reference unit to a finer one (10x10 grid). As the reviewer has pointed out, replicating this analysis to other temporal or spatial scales would be easy. Regarding the regions of analysis, we have used these three regions (Northwest, Hinterland and Mediterranean) because we want to know the overall behaviour of trends. We coincide with the reviewer in that it might not be the

most appropriate partition, since their mean values or dynamics are not homogenous. For this reason we have lowered the scale to the NUTS3. In this sense, this second stratification has been chosen because we tried to increase the degree of detail in the trends description within each region. In any case, note that those regions are used in other studies that we took as reference to stablish comparisons; an all official statistics in Spain are referred to them.

REVIEWER 2: Some descriptive statistics of the fire database in the 2.2 Fire data section would probably help to justify the spatial and temporal stratification used.

AUTHORS: This is a very good suggestion. Indeed we should have already provided this. We have added a table with this information in the Fire data section.

REVIEWER 2: Lines after 185 explain three algorithms for change point detection. Why settings were determined to find at least one, but no more than two breakpoints in PELT, and one (Q=1) in BinSeg? This makes sense for comparison purposes with AMOC and Pettitt, but is there not a risk to miss other significant changes?

AUTHORS: We didn't limit change point detection to 1. It is true that AMOC and Pettit methods only are able to detect 1 point, but PELT generally reports more than one. The thing is that most of the time there is only 1 point detected, although in those cases where more than one was detected we reported them (Table 1). However, we would definitively prioritize the most coincident change point as the most likely or strongest one among all methods.

REVIEWER 2: The authors refer to CCAA in Spain the international readers will not be familiar with, i.e. Andalusia, Galicia or Asturias, not in Figure 1. Labels seem to be missing. What is the black line crossing the land cover map?

AUTHORS: We agree that it will be more useful to include place-names of provinces (NUTS3) and CCAA (NUTS2), for this reason we have finally included a politic map of Spain in the new version of Figure 1. On the other hand, the black line crossing the land

cover map represented the limit between both biogeographic regions (Eurosiberian and Mediterranean). However, we finally removed it because we believe that land cover alone describes best our study area.

REVIEWER 2: Regarding Figures 4 and 5, Sen's slope values are hard to distinguish.

AUTHORS: We really appreciate this observation, thus in Figures 4 and 5 we have changed the continuous colour scale to a discrete colour scale for the variables mapped.

REVIEWER 2: Why is the level for correlation in table 3 set to 0.43? Please explain.

AUTHORS: This threshold was established based on the actual values we retrieved from PCA-Varimax. There is no rule-of-thumb when it comes to determine a correlation threshold. We now realise that reporting a cut-off value of 0.43 it's rather awkward. In fact, the actual value is 0.4 but, again, this is based on the two most correlated featured in each component.

---

## Referee Comment (RC3) · Anonymous Referee #3 · 24 Aug 2017

The manuscript analyses the dynamics of fire regime components, more accurately fire regime elements, as two crucial fire regime metrics are not addressed - fire frequency (see specific comment below) and fire severity – for peninsular Spain. Results generally concur with the findings of previous studies, namely Moreno et al. (2014) for Spain and Turco et al. (2016) for southern Europe. Hence the novelty resides mostly in examining the trends in variables other than number of fires and area burned.

On the methods side I commend the authors on the depth and diversity of the statistical analysis, which I believe has not been seen before in similar studies.

I see improvement opportunities on the Discussion section, which is comparatively weaker. What are the motives behind the trends found? Given the existence of previous analysis of this type one would expect a deeper perception/development of the

discussion on the driving causes, be it fire weather, land management, or fire management. There are reasons to believe that the major influence is/has been the extraordinary investment (perhaps the highest in the world) that Spain has devoted to fire suppression, see Seijo & Gray (2012). For comparison the authors can check Fernandes et al. (2014), which examined trends in northern Portugal where a shift towards decreasing area burned did not happen in 1980s-1990s, presumably because of unsuccessful/insufficient firefighting efforts.

Another aspect in need of improvement is a joint explanation of the trends, i.e. an attempt to relate trends detected for the different metrics can be made. E.g. in NW Spain, large fires have increased, there are more winter fires, and summer burned area did decrease. It is likely that the fire exclusion policy in place is resulting in less area burned. Because of the repression of fire use to manage land, people will be inclined to use fire in winter (when fire preparedness is low) rather than on summer, but this traditional use of fire will not have an impact on the extent of flammable landscapes, because fires are usually small. Hence a more flammable landscape is developing, explaining the increase in the number of large fires, particularly in years with more extreme fire weather days and/or higher number of extreme fire weather days, a consequence of climate change. This is the type of inference/analysis that would really benefit the ms.

I also advise the authors on doing some discussion regarding the limitations of the trend analysis methods. Because fires are self-limiting the landscape preserves a memory of fire, especially where fires are larger or fire frequency is higher. Thus, what the analysis reveals as decreasing trends may in fact be a consequence of relatively long fire cycles in relation with landscape-level fuel build-up, and this may really impact the results.

References:

Seijo, F., Gray, R., 2012. Pre-industrial anthropogenic fire regimes in transition: the case of Spain and its implications for fire governance in Mediterranean type biomes.

Hum. Ecol. Rev. 19, 59e69. Fernandes, P.M., Loureiro, C., Guiomar, N., Pezzatti, G.B., Manso, F., Lopes, L. 2014. The dynamics and drivers of fuel and fire in the Portuguese public forest. J. Environ. Manage. 146, 373-382.

Specific comments

P1, L24. Replace "conversely": it has the opposite meaning of what you are trying to convey. P2, L36. Vegetation type and structure, as variation in fire behaviour is high within a given vegetation type. P2, L41. "improve", not "improving". P2, L41. Rephrase. "How wildfire works" is quite subjective in its meaning. P2, L50. remains. P2, L68. This sentence lacks a 2nd part: "Since most studies focus mainly on analysing 'generic' fire (number of fires and burned area)." P2, L88-95, L96-98. Too much detail here on the methods used. Delete or reduce substantially. P3, L103. Environmental can be understood as incorporating some of the climatic and topographic features. Replace by land cover, or vegetation, or fuel. P3, L109. I don't think ash (Fraxinus) is a relevant land cover type. This region also has a quite important component of forest plantations such as Pinus radiata and eucalypts. P4, L114-115. Add other important oak (Q. suber) and pine (P. nigra, pinaster, sylvestris) species. P4, L144. By definition "fire frequency" is the number of times a given area has burned in the past, divided by the number of years considered, thus an annual probability. You must rename this variable for what it really is, i.e. Number of fires, here and elsewhere in the text and figures. P4, L144-145. Regardless of size.
* * *

---

## Author Comment (AC3) · 4 Sep 2017

REVIEWER 3: The manuscript analyses the dynamics of fire regime components, more accurately fire regime elements, as two crucial fire regime metrics are not addressed - fire frequency (see specific comment below) and fire severity – for peninsular Spain. Results generally concur with the findings of previous studies, namely Moreno et al. (2014) for Spain and Turco et al. (2016) for southern Europe. Hence the novelty resides mostly in examining the trends in variables other than number of fires and area burned. On the methods side I commend the authors on the depth and diversity of the statistical analysis, which I believe has not been seen before in similar studies. I see improvement opportunities on the Discussion section, which is comparatively weaker. What are the motives behind the trends found? Given the existence

of previous analysis of this type one would expect a deeper perception/development of the discussion on the driving causes, be it fire weather, land management, or fire management. There are reasons to believe that the major influence is/has been the extraordinary investment (perhaps the highest in the world) that Spain has devoted to fire suppression, see Seijo & Gray (2012). For comparison the authors can check Fernandes et al. (2014), which examined trends in northern Portugal where a shift towards decreasing area burned did not happen in 1980s-1990s, presumably because of unsuccessful/insufficient firefighting efforts. Another aspect in need of improvement is a joint explanation of the trends, i.e. an attempt to relate trends detected for the different metrics can be made. E.g. in NW Spain, large fires have increased, there are more winter fires, and summer burned area did decrease. It is likely that the fire exclusion policy in place is resulting in less area burned. Because of the repression of fire use to manage land, people will be inclined to use fire in winter (when fire preparedness is low) rather than on summer, but this traditional use of fire will not have an impact on the extent of flammable landscapes, because fires are usually small. Hence a more flammable landscape is developing, explaining the increase in the number of large fires, particularly in years with more extreme fire weather days and/or higher number of extreme fire weather days, a consequence of climate change. This is the type of inference/analysis that would really benefit the ms.

AUTHORS: Firstly, we would like to thank the reviewer for his/her useful comments and suggestions about the manuscript. Regarding to not include fire severity and fuels, we understand that both factors are remarkable important in fire research. Unfortunately, these aspects are not possible to be addressed due to several of factors. First of all, there is no information about fire severity. At best, severity can be derived from remote sensing imagery, but that's not feasible given both the spatial and temporal scales of analysis. On the other hand, finding direct causes for each trend detected is beyond the scope of our work. We have tried to mention certain factors that might be behind the detected changes and trends. Regarding the novelty of our work, indeed this topic is not new in the context of Spain we believe that our work goes a step further in

providing insights and analyze dynamics in fire regime features. Specifically we have (i) extended the analysis to other fire regime features in our change-point analysis; (ii) we also apply traditional trend analysis to these other features; (iii) assessing not only the sign of trends, but its magnitude, which has not yet been addressed; (iv) at different scales; (v) this work would also allow progress in the fire regimes zoning; (vi) finally, our most novel contribution is exploring the relationships and association among trends in fire features using Principal Components in an effort to provide a more synthetic interpretation as well. We also provide a summary map of the main trends detected which allows outlining homogeneous zones of temporal dynamics at province level. To our knowledge such kind of analysis and cartographic outputs has not yet been done.

REVIEWER 3: I also advise the authors on doing some discussion regarding the limitations of the trend analysis methods. Because fires are self-limiting the landscape preserves a memory of fire, especially where fires are larger or fire frequency is higher. Thus, what the analysis reveals as decreasing trends may in fact be a consequence of relatively long fire cycles in relation with landscape-level fuel build-up, and this may really impact the results. References: Seijo, F., Gray, R., 2012. Pre-industrial anthropogenic fire regimes in transition: the case of Spain and its implications for fire governance in Mediterranean type biomes. Hum. Ecol. Rev. 19, 59e69. Fernandes, P.M., Loureiro, C., Guiomar, N., Pezzatti, G.B., Manso, F., Lopes, L. 2014. The dynamics and drivers of fuel and fire in the Portuguese public forest. J. Environ. Manage. 146, 373-382.

AUTHORS: We really appreciate this observation regarding to the limitations of the trends analysis, since we only have commented benchmark concerning with the spatial units employed or the necessary exploration of deeper insights causes. Thus, we have finally included this comment in the discussion section.

Specific comments REVIEWER 3: P1, L24. Replace "conversely": it has the opposite meaning of what you are trying to convey.

AUTHORS: We have replaced the "conversely" to "similarly".

REVIEWER 3: P2, L36. Vegetation type and structure, as variation in fire behaviour is high within a given vegetation type.

AUTHORS: We really appreciate this observation and we have included in the corresponding sentence.

REVIEWER 3: P2, L41. "improve", not "improving".

AUTHORS: We have changed "improving" to "improve".

REVIEWER 3: P2, L41. Rephrase. "How wildfire works" is quite subjective in its meaning.

AUTHORS: We appreciate this useful observation and we have to tried to make this part of the sentence more objective.

REVIEWER 3: P2, L50. remains.

AUTHORS: We have corrected this word.

REVIEWER 3: P2, L68. This sentence lacks a 2nd part: "Since most studies focus mainly on analysing 'generic' fire (number of fires and burned area)."

AUTHORS: We have changed the beginning of this sentence to complete the idea.

REVIEWER 3: P2, L88-95, L96-98. Too much detail here on the methods used. Delete or reduce substantially.

AUTHORS: We have reduced the length of this paragraph to only introduce general aspects of the methodology used.

REVIEWER 3: P3, L103. Environmental can be understood as incorporating some of the climatic and topographic features. Replace by land cover, or vegetation, or fuel.

AUTHORS: We have changed this concept to "vegetation communities".

REVIEWER 3: P3, L109. I don't think ash (Fraxinus) is a relevant land cover type. This region also has a quite important component of forest plantations such as Pinus radiata and eucalypts.

AUTHORS: We really appreciate this particular nuance of this region and we have finally considered including the forest plantations component.

REVIEWER 3: P4, L114-115. Add other important oak (Q. suber) and pine (P. nigra, pinaster, sylvestris) species.

AUTHORS: We have included these species in this section.

REVIEWER 3: P4, L144. By definition "fire frequency" is the number of times a given area has burned in the past, divided by the number of years considered, thus an annual probability. You must rename this variable for what it really is, i.e. Number of fires, here and elsewhere in the text and figures.

AUTHORS: We appreciate the observation made by the reviewer concerning to "fire frequency". Fire frequency is replaced by number of fires as defined in the Glossary of Wildland Fire Terminology of the National Wildfire Coordinating Group in 2008. Although we used the term frequency in accordance with the classic FAO 1986 terminology (FAO (1986) Wildland Fire Management Terminology. FAO Forestry Paper 70, Food and Agriculture Organization of the United Nations, Rome/ http://www.fao.org/docrep/016/ap456t/ap456t00.pdf), widely accepted in Spain.

REVIEWER 3: P4, L144-145. Regardless of size.

AUTHORS: We have corrected these mistakes.